# Evaluation of the Severity of Major Depression Using a Voice Index for Emotional Arousal

**DOI:** 10.3390/s20185041

**Published:** 2020-09-04

**Authors:** Shuji Shinohara, Hiroyuki Toda, Mitsuteru Nakamura, Yasuhiro Omiya, Masakazu Higuchi, Takeshi Takano, Taku Saito, Masaaki Tanichi, Shuken Boku, Shunji Mitsuyoshi, Mirai So, Aihide Yoshino, Shinichi Tokuno

**Affiliations:** 1Department of Bioengineering, Graduate School of Engineering, The University of Tokyo, 7-3-1 Hongo, Bunkyo-ku, Tokyo 113-8656, Japan; nakamura@bioeng.t.u-tokyo.ac.jp (M.N.); higuchi@bioeng.t.u-tokyo.ac.jp (M.H.); mitsuyoshi@bioeng.t.u-tokyo.ac.jp (S.M.); tokuno@bioeng.t.u-tokyo.ac.jp (S.T.); 2Department of Psychiatry, National Defense Medical College, 3-2 Namiki, Tokorozawa, Saitama 359-8513, Japan; toda1973@ndmc.ac.jp (H.T.); tsaito@ndmc.ac.jp (T.S.); mtanichi@gmail.com (M.T.); aihide@ndmc.ac.jp (A.Y.); 3PST Inc., Industry & Trade Center Building 905, 2 Yamashita-cho, Naka-ku, Yokohama, Kanagawa 231-0023, Japan; omiya@medical-pst.com (Y.O.); takano@medical-pst.com (T.T.); 4Department of Neuropsychiatry, Faculty of Life Sciences, Kumamoto University, 1-1-1 Honjo, Chuo-ku, Kumamoto, Kumamoto 860-8556, Japan; shuboku@gmail.com; 5Department of Psychiatry, Tokyo Dental College, 2-9-18, Misakicho, Chiyoda-ku, Tokyo 101-0061, Japan; somirai@tdc.ac.jp

**Keywords:** arousal level, emotion, major depression severity, voice index, Hurst exponent, zero-crossing rate, Hamilton Rating Scale for Depression

## Abstract

Recently, the relationship between emotional arousal and depression has been studied. Focusing on this relationship, we first developed an arousal level voice index (ALVI) to measure arousal levels using the Interactive Emotional Dyadic Motion Capture database. Then, we calculated ALVI from the voices of depressed patients from two hospitals (Ginza Taimei Clinic (H1) and National Defense Medical College hospital (H2)) and compared them with the severity of depression as measured by the Hamilton Rating Scale for Depression (HAM-D). Depending on the HAM-D score, the datasets were classified into a no depression (HAM-D < 8) and a depression group (HAM-D ≥ 8) for each hospital. A comparison of the mean ALVI between the groups was performed using the Wilcoxon rank-sum test and a significant difference at the level of 10% (*p* = 0.094) at H1 and 1% (*p* = 0.0038) at H2 was determined. The area under the curve (AUC) of the receiver operating characteristic was 0.66 when categorizing between the two groups for H1, and the AUC for H2 was 0.70. The relationship between arousal level and depression severity was indirectly suggested via the ALVI.

## 1. Introduction

Globally, economic loss due to mental health disorders is an issue that requires immediate and appropriate attention [1,2]. In addition, early detection of depression can help to reduce the number of suicides caused by depression and prevent infanticide by mothers with postpartum depression (PPD) [3]. To address these issues, an easily accessible and low-cost mental health screening method is required. The primary mental health assessment tools currently in use include medical interviews by specialists (e.g., the Hamilton Rating Scale for Depression (HAM-D)) [4], self-administered questionnaires (e.g., the general health questionnaire [5] and the Beck Depression Inventory [6]). However, the practicality of medical interviews by specialists is limited by the restricted number of patients that can be examined, and self-administered questionnaires have reporting bias issues [7]. Reporting bias refers to participants’ intentional disclosure or suppression of certain information (e.g., medical history, smoking history). PPD is usually identified using self-report measures; therefore, it is possible that mothers with PPD are unwilling to report it because of social desirability bias [3]. Further, assessment tools using biomarkers such as saliva and blood have also been studied [8,9,10,11]. For example, Hori et al. [11] clarified that the ribosomal proteins L17 and L34 play a role in depression and stress-vulnerability, in which their gene expression levels in the blood can serve as diagnostic markers. However, these tools have drawbacks concerning cost and patient burden.

Meanwhile, with the increasing usage of smartphones, a pathological analysis using speech data has attracted attention [12,13,14,15,16]. According to long-established findings, patients with depression have unique speech characteristics [17] and listeners can perceive their distinctive prosody [18,19]. Moreover, speech analysis using smartphones is non-invasive and can be conveniently and remotely accessed without any dedicated device. In addition, voice monitoring has the advantage of being more sensitive to daily changes than self-administered questionnaires.

Many recent studies denote that speech characteristics are an effective predictor of the signs and severity of depression [20]. For example, Cannizzaro et al. [21] examined the interdependence between the HAM-D and voice acoustics and found a strong correlation between the HAM-D and speaking rate or pitch variation. Speaking rate was calculated by dividing the number of syllables spoken by the length of the sample measured in seconds. For each sample, pitch variation was extracted using the Kay Elemetrics Computerized Speech Laboratory 4400 (Kay Elemetrics Corporation, NJ, USA) pitch contour analysis. The correlation coefficient between the speaking rate and the HAM-D score was -0.89, while its value was -0.74 between pitch variation and the HAM-D score.

Faurholt-Jepsen et al. [22] found that depressive symptoms, which were measured by the HAM-D, and manic symptoms, which were examined through the Young Mania Rating Scale (YMRS) [23], can be classified accurately via audio signal features for patients with bipolar disorder. They used the open-source Media Interpretation by Large feature-space Extraction toolkit (openSMILE) [24] to extract 6552 voice features from the speech and used random forest to classify the three emotional states (depressive state, euthymic state, manic or mixed state) of the patients. The area under the curve (AUC) of the receiver operating characteristic (ROC) was 0.78 when categorizing between the depressive and euthymic states.

Hashim et al. [25] extracted 19 voice features related to spectrum-based measures, such as power spectral density (PSD) and Mel-Frequency Cepstral Coefficient (MFCC), and 48 voice features related to timing-based measures, such as interval probability density functions that estimated from histograms based on the voiced/unvoiced/silence labeling of the frames. These 67 voice features were used as explanatory variables, while HAM-D scores were used as objective variables to create a prediction formula using multiple regression analysis. As a result, the mean absolute error between the predicted HAM-D scores and the actual HAM-D scores was approximately 2.

Furthermore, Gabrieli et al. [3] investigated whether there is a significant difference in the acoustical properties of the vocalizations of infants between depressed and healthy mothers. They extracted acoustic features (fundamental frequency, first four formants, and intensity) from recordings of crying infants. The trained model showed that acoustical features can be successfully used to identify mothers with PPD with high accuracy (89.5%).

Moreover, Yang et al. [19] indicated that changes in the severity of depression, measured by the HAM-D, can be deciphered through a switching pause. They specify switching pause as “the pause duration between the end of one speaker’s utterance and the start of an utterance by the other” [19] (p. 145). Additionally, Taguchi et al. [26] verified that MFCC 2 (i.e., the second dimension of the MFCC) is efficient in classifying between depressive and healthy individuals. The AUC of MFCC 2 was 0.88 when classifying between patients with depression and healthy individuals. 

Although Faurholt-Jepsen et al. [22] indicated a relationship between the HAM-D score and voice index, the issue of overfitting may remain because of the many audio signal features extracted from openSMILE [24]. In contrast, Taguchi et al. [26] showed that healthy individuals and patients with depression could be grouped based only on MFCC 2; however, MFCC 2 did not correlate with the severity of depression. 

With the progress of brain science research, the correlation between depression and arousal has been attracting attention in recent years, and many studies using physiological indices, such as electroencephalogram (e.g., late positive potential (LPP) amplitude), magnetoencephalography (e.g., neuromagnetic oscillatory activity), and skin conductance magnitude, have been conducted [27,28,29]. Additionally, the methods for assessing emotional arousal from voice have been studied, and the relationship between arousal level, vocal intensity, and pitch has been validated [30,31].

It is known that stress affects emotions [32]. In addition, major depression is characterized by episodes such as diminishment of interest or pleasure and feeling of sadness and emptiness in the DSM-IV-TR (Diagnostic and Statistical Manual of Mental Disorders, Fourth Edition, Text Revision) [33]. From a voice perspective, the voices of patients with depression have been reported to be dull, monotonous, and lifeless [17]. Meanwhile, using the Interactive Emotional Dyadic Motion Capture (IEMOCAP) database [34], Shinohara et al. [35] showed that individuals feel a high level of arousal against voices characterized by anger, excitement, and happiness, and a low level of arousal against voices characterized by neutrality, sadness, and disgust. Through research using physiological indicators such as LPP amplitude and skin conductance, Benning and Ait Oumeziane [29] suggested that underarousal and low positive emotion might be the core emotional components of subclinical depression. From these findings, we hypothesized that the voices of depressed patients tended to have a low level of arousal. However, in the early stages of depression, excessive stress may cause hyperarousal.

In this paper, we examined the possibility of detecting depression using a voice index based on the estimation of speech’s arousal level. First, we constructed an arousal level voice index (ALVI) for calculating arousal levels from voices by integrating emotional voices in IEMOCAP. We focused on the relationship between the Hurst exponent (HE) and the zero-crossing rate (ZCR) of the speech waveform when formulating ALVI. Next, we used the algorithm to determine the arousal level from the voice of patients with depression and investigated the relationship between the arousal level and the severity of depression according to HAM-D scores.

## 2. Materials and Methods

### 2.1. Acquisition of Data

#### 2.1.1. Data about Arousal Level

We used the IEMOCAP database [34] to create an algorithm that calculates the arousal level from the acoustic features of a person’s voice. The database contains audio recordings of dyadic mixed-gender pairs of actors. There were five sessions in total, that is, the voices of 10 actors were aggregated and then manually divided into utterance units. Notably, the arousal level of each utterance was gauged by two different annotators on a 5-point scale. Further, the arousal level of each utterance was equal to the average evaluation values given by each evaluator. The Cronbach alpha coefficient was computed to test the reliability of the evaluations between the two raters. The coefficient was 0.607 [34]. In total, 10039 utterances were included in this study, and among these utterances, low arousal data with arousal level of 2 or less (n = 1112, mean ± SD = 1.92 ± 0.19) and high arousal data of level 4 or more (n = 1692, mean ± SD = 4.19 ± 0.28) were used.

#### 2.1.2. Data about the Severity of Depression

The patients’ speech data were collected from outpatients with major depressive disorders after acquiring informed consent from participants attending two hospitals, Ginza Taimei Clinic (H1) and National Defense Medical College Hospital (H2). The patients were directed to read the 17 Japanese phrases in a fixed order. However, all 17 phrases collected at both hospitals were not the same; 10 among them were common at both hospitals. Therefore, this study elicited the 10 common phrases and Table 1 illustrates them accordingly. Additionally, the speech of the two groups was recorded in a quiet and controlled environment and the voice was recorded using a pin microphone (ME52W, Olympus, Tokyo, Japan) placed on the chest about 15 cm from the mouth. The recording equipment was a portable recorder R-26 (Roland, Shizuoka, Japan) and the record format was linear PCM with the sampling frequency and the number of quantization bits at 11,025 Hz and 16, respectively. 

In addition to voice recording, the HAM-D was also used by psychiatrists to appraise the severity of major depression. For each speech data acquisition session, HAM-D data were obtained from each participant and paired with the recorded speech data. Table 2 shows participants’ information at each hospital. 

Patients were included if they had been diagnosed with a major depressive disorder according to the DSM-IV-TR [33] and were over the age of 20 years. However, the participants were excluded if the presence of serious physical disorders or organic brain disease was confirmed. Specifically, the Mini-International Neuropsychiatric Interview [36] was employed by a psychiatrist to evaluate these conditions.

The protocol of this study was designed as per the Declaration of Helsinki and relevant domestic guidelines issued by the concerned authority in Japan. The protocol was approved by the ethics committee of the faculty of medicine, The University of Tokyo (no. 11572) and the ethics committee of National Defense Medical College (no. 2248).

According to Japanese law, the sensitivity of audio files is similar to that of any other personal information and cannot be published without consent. In this research protocol, we did not obtain consent from the subjects to publish the raw audio files as a corpus. Hence, we only published the analysis data and the source code of the analysis program.

### 2.2. Proposed Method

This study proposes a new voice index for emotional arousal level calculated based on the relationship between two scales, namely HE and ZCR.

HE is an index that is regularly used in stock price analyses and, hence, it is widely used as a method to measure the characteristics of long-term memory [37]. The following equation was used to obtain the standard deviation (SD) σ(τ) of the difference Δx(t,τ) between signal x(t) at time t and signal x(t+τ) after τ elapsed (where x is the raw audio signal data that reflects the sound pressure level) (Equation (1)).
(1)$$\begin{array}{l} \sigma \left(\tau \right)=\sqrt{\frac{\sum_{t=1}^{n-\tau }{\left(\Delta x\left(t,\tau \right)-\bar{\Delta x\left(\tau \right)}\right)}^{2}}{n-\tau -1}}\\ \Delta x\left(t,\tau \right)=x\left(t+\tau \right)-x\left(t\right)\\ \bar{\Delta x\left(\tau \right)}=\frac{\sum_{t=1}^{n}\Delta x\left(t,\tau \right)}{n} \end{array}$$

It is known that there is a power law relationship (i.e., $\sigma \left(\tau \right)\propto \tau^{\alpha }$ or log(σ(τ))∝αlog(τ)) between σ(τ) and τ when τ is a small value. Moreover, α is called an HE and in this research, the range of τ was defined as τ≤1.46 ms (16 data points). 

The specific calculation procedure of HE is as follows: first, the value of τ is changed from τ=1/11,025 Hz to τ=16/11,025 Hz, where each value of σ(τ) is calculated using Equation (1); next, a regression line is drawn for the 16 points (log(τ), log(σ(τ))); the slope of this regression line is defined as HE. However, if the coefficient of determination when calculating the regression line between log(σ(τ)) and log(τ) is less than 0.9, it is considered that the power law relationship does not last, and the value of HE is returned as invalid. Thus, HE is an index of displacement from an initial position over time and theoretically, HE = 0.0 is in white noise and HE = 0.5 is in brown noise. HE is represented by HE = 2-D where D is the fractal dimension. Therefore, HE can be regarded as an index representing waveform smoothness (i.e., the opposite of fractal dimension). 

Meanwhile, ZCR is an index often used in studies on voice activity detection and classification of voiced/unvoiced sounds, indicating the rate at which the signal changes from positive to negative, or conversely, from negative to positive [38,39,40]. In summary, the rate at which the signal crosses the reference line is denoted. Specifically, the ZCR of signal x(t), which is a signal of length n, is expressed using the following equation (Equation (2)):(2)$$\begin{array}{l} ZCR=\frac{\sum_{t=1}^{n-1}I\left(y\left(t\right)y\left(t+1\right)\right)}{n-1}\\ y\left(t\right)=x\left(t\right)-\bar{x}\\ \bar{x}=\frac{\sum_{t=1}^{n}x\left(t\right)}{n}\\ I\left(y\left(t\right)y\left(t+1\right)\right)=\{\begin{array}{cc} 1 & if\;y\left(t\right)y\left(t+1\right)<0\\ 0 & else \end{array} \end{array}$$
$\bar{x}\;$represents the time average of x(t). y(t) represents a value obtained by correcting x(t) so that $\bar{x}\;$becomes 0.

Notably, the relationship between HE and ZCR as a voice index of arousal was examined. First, each speech phrase was separated into frames of length L where individual frame lengths were set with an overlap of L/4. In this study, L was set to 46.44 ms (i.e., 512 data points). Second, HE and ZCR were calculated for each frame; *n* was set to 512 in Equations (1) and (2). Then, among the frames contained in the utterance, the frames with valid HE values were selected. The mean value of HE of those frames was set as the HE of the utterance, and the same was applied to ZCR.

Figure 1 shows a scatter plot between HE and ZCR calculated from the utterances in the IEMOCAP database. We performed a logistic regression analysis wherein HE and ZCR were used as explanatory variables. Alternatively, data with an arousal level of 2 or less were set to 0, and data of 4 or more were set to 1. These binary values (0, 1) were used as levels of an objective variable, and the discrimination score was termed the arousal level voice index (ALVI) expressed by the following formula.
(3)$$ALVI=\frac{1}{1+\exp\left[-\left(21.60HE+116.37ZCR-22.39\right)\right]}$$

ALVI takes values in the range (0.0, 1.0). Figure 1 also shows a straight line 21.60HE+116.37ZCR−22.39=0 that aligns a set of points where ALVI = 0.5. The mean ALVI values for the low and high arousal groups were 0.33 ± 0.26 and 0.78 ± 0.22, respectively. Figure 2 shows the receiver operating characteristic (ROC) curve when the utterance data of low and high arousal levels are classified by ALVI.

The area under the curve (AUC) of the ROC was 0.89. The cut-off value was ALVI = 0.53 and the sensitivity and specificity at the time were 0.86 and 0.78, respectively. The above analysis was performed using the statistical software R version 3.6.1 (2019-07-05) [41]. We used the R packages of Epi version 2.41 for AUC calculation, and Car version 3.0.8. for the two-way analysis of variance (ANOVA). The operating system used was Windows 10. The following analysis was also performed using R unless otherwise specified.

No filtering or noise reduction was performed on the preprocessing of the audio signal. However, the extraction of each phrase from each patient’s voice recording was done manually, and the silent sections between phrases were manually deleted.

## 3. Results

### 3.1. HAM-D Score

Although there are various arguments about severity classification using the HAM-D [42], by using Hashim’s method in this study, patients’ data were divided into two groups based on HAM-D scores: a no depression group with a HAM-D score of <8 and a depression group with a HAM-D score of ≥8 [25]. Table 3 displays the mean HAM-D for each group per hospital.

As a result of the Wilcoxon rank-sum test, significant differences in the HAM-D score were observed between the groups at H1 (*p* = 1.22 × 10^−8^). Similarly, there was a significant difference at H2 as well (*p* = 1.56 × 10^−13^).

### 3.2. Performance Evaluation of ALVI

In this section, the results of applying ALVI to the voice of patients with depression are outlined. Figure 3 shows a scatter diagram of HE and ZCR calculated from each utterance (n = 1780) of depressed patients, and a straight line 21.60HE+116.37ZCR−22.39=0 is also shown, as in Figure 2.

Depressed patient voices were collected at two hospitals, H1 and H2. As shown in Table 2, the age groups of the patients are quite different. Likewise, the sound field environment may be different because of the difference in size and shape of the examination room in both hospitals. In addition, because the soundproofing conditions in the examination rooms were different, there may be differences in the noise mixed in the recorded data. Therefore, an analysis has been performed for each hospital.

Figure 4a shows the mean ALVI for each group per hospital. However, the ALVI value of each patient was the mean value of the ALVI of each phrase. At H1, the ALVI mean values of the no depression group and depression group were 0.25 ± 0.19 (n = 10) and 0.14 ± 0.10 (n = 78), respectively. At H2, the ALVI mean values of the no depression group and depression group were 0.41 ± 0.18 (n = 65) and 0.28 ± 0.17 (n = 25), respectively. Figure 4b shows the mean HAM-D score for each group per hospital, which is also shown in Table 3. According to Figure 4, the ALVIs of H2 are totally higher than that of H1. On the contrary, the mean HAM-D scores of each group in H1 is higher than that of H2.

A comparison between the groups was performed using the Wilcoxon rank-sum test, and it determined a significant difference at the level of 10% (*p* = 0.094) at H1 and 1% (*p* = 0.0038) at H2. The AUC, at the moment when the groups were identified using ALVI, was 0.66 for H1 (cutoff point = 0.20, sensitivity = 0.60, specificity= 0.78) and 0.70 for H2 (cutoff point = 0.23, sensitivity = 0.85, specificity = 0.56). 

Next, we inspected the effect of different phrases on the ALVI for each hospital. Two-way ANOVA of ALVI was conducted for two factors, groups (i.e., no depression or depression) and phrases (i.e., 10 phrases). At H1, significant differences in both group and phrase factors (F(1, 860) = 33.52, *p* = 9.87 × 10^−9^, F(9,860) = 28.26, *p* < 2.00 × 10^−16^) were noted; however, there was no interaction between them (F (9, 860) = 0.375, *p* = 0.95). Similarly, at H2, there were significant differences in the factors of both groups and phrases (F (1, 880) = 57.09, *p* = 1.04 × 10^−13^, F(9, 880) = 16.56, *p* < 2.00 × 10^−16^), and no interaction was observed again (F(9, 880) = 0.488, *p* = 0.88). Figure 5 shows the mean ALVI for each phrase in the no depression and depression groups. Finally, in all phrases, the ALVI of the no depression group was higher than that of the depression group in both hospitals. 

Table 4 highlights the classification performance between the no depression and depression groups through ALVI. The table shows the results of comparing the mean ALVI of each group using the Wilcoxon rank-sum test and the AUC for each phrase. At H1, the discrimination performance for phrase 12 was the highest, and the AUC was 0.70 (cutoff point = 0.14, sensitivity = 0.70, specificity = 0.73). In contrast, at H2, the discrimination performance for phrase 5 was the highest, and the AUC was 0.74 (cutoff point = 0.30, sensitivity = 0.85, specificity = 0.60).

## 4. Discussion

First, we focused on the relationship between the HE and the ZCR and developed a voice index (i.e., ALVI) for arousal. Next, ALVI was derived from the voices of depressed patients and applied for the identification of severity-based groups. Consequently, the ALVI of the no depression group was significantly higher than that of the depression group. In other words, the association between arousal level and depression severity was indirectly suggested via the voice index ALVI. However, the arousal level used in the development of ALVI was an evaluation value given by the annotator, and not based on the subject’s evaluation. Hence, in the future, it is imperative to investigate the relationship between ALVI and physiological indicators.

The AUC of ALVI was about 0.7 in the highest phrase. Comparably, Taguchi et al. [26] presented the AUC of MFCC 2 as 0.88 when classifying between patients with depression and healthy individuals. In our study, we did not compare healthy individuals with depressed patients as in Taguchi et al. Instead, we compared two patient groups. This may be the reason why the AUC of our ALVI became lower. We plan to compare healthy individuals with depressed patients in future studies.

Regarding the severity of depression, the model classifying a depressive state versus a euthymic state had an AUC of 0.78 in the study of Faurholt-Jepsen et al. [22]. Here, a depressive state is defined by a HAM-D score ≥13 and YMRS score <13, while a euthymic state is defined as a HAM-D score <13 and YMRS score <13. Even though no simple comparison is viable, the AUC shown here is higher than that of ALVI. However, it may be advantageous because the ALVI is unlikely to be overfitted. After all, it consists of only two indices, HE and ZCR. 

Moreover, the voices of patients with depression have been reported to be dull, monotonous, and lifeless based on a qualitative point of view [17]. As seen in Equation (3), ALVI increases as both HE and ZCR increase. Notably, HE is represented by HE = 2-D, where D is the fractal dimension. Therefore, HE can be regarded as an index representing waveform smoothness (i.e., the opposite of fractal dimension). Plus, as described in the definition, ZCR is higher in white noise and lower in smooth waveforms. Therefore, there may be a negative relationship between HE and ZCR. 

In general, unvoiced sounds, such as “s”, have a higher frequency and ZCR. Previous findings have shown that voices with lower short-time energy and higher ZCR are likely to be unvoiced sounds [38,39,40]. Furthermore, Shinohara et al. [43] proved that the pitch detection rate of patients with depression was lower than that of healthy individuals, where the pitch detection rate can also be defined as the proportion of voice sounds. Perhaps a sound may tend to become an unvoiced sound not only when ZCR is high, but also when HE is simultaneously low. 

This study has limitations. First, all participants had to use the same phrases for accurate evaluation because the current ALVI values vary depending on the phrase. In addition, they had to read the phrases in a fixed order from P1 to P17. Our future task is to apply ALVI to spontaneous speech. Future studies should explore the reasons for such differences among phrases. 

Second, the sample size of the group was small because we collected voices from two hospitals, H1 and H2, where the age groups were very different. The reason for this may be that H1 is located in the city center and most of the patients are young workers who commute to the city center, while H2 is located in the suburbs where many retired local residents live. Depression severity was uneven at both hospitals. The reason for this may be that H2 is a university hospital and there are many patients that have been treated in another hospital. Therefore, future studies should include a bigger sample acquired in the same environment.

Third, there was a difference in ALVI values due to the different recording environments of the two hospitals. To carry out a universal screening in the future, it is necessary to eliminate environmental dependence. In this study, we used no pre-processing such as noise reduction and filtering. Thus, in the future, we would study pre-processing methods including calibration.

Fourth, we recorded audio using a pin microphone in this study. In the future, it is necessary to verify whether the same tendency as in this research can be seen using smartphones and Internet of Things devices.

By monitoring the severity of depression using ALVI daily, we can encourage people to visit a hospital before they become depressed or during the early stages of depression. This may lead to decrease in suicides caused by depression and reduced economic loss due to treatment costs and interference with work.

## Figures and Tables

**Figure 1 sensors-20-05041-f001:**
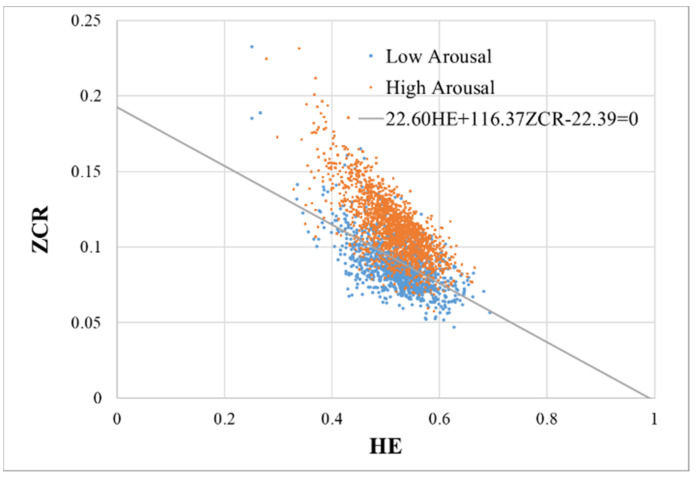
A scatter plot between Hurst exponent and zero-crossing rate calculated from the utterances in the Interactive Emotional Dyadic Motion Capture database. Each data point represents data for each utterance. The low arousal level data are shown in blue and the high arousal level data are shown in orange. (Note. ZCR: zero-crossing rate; HE: Hurst exponent; IEMOCAP: Interactive Emotional Dyadic Motion Capture).

**Figure 2 sensors-20-05041-f002:**
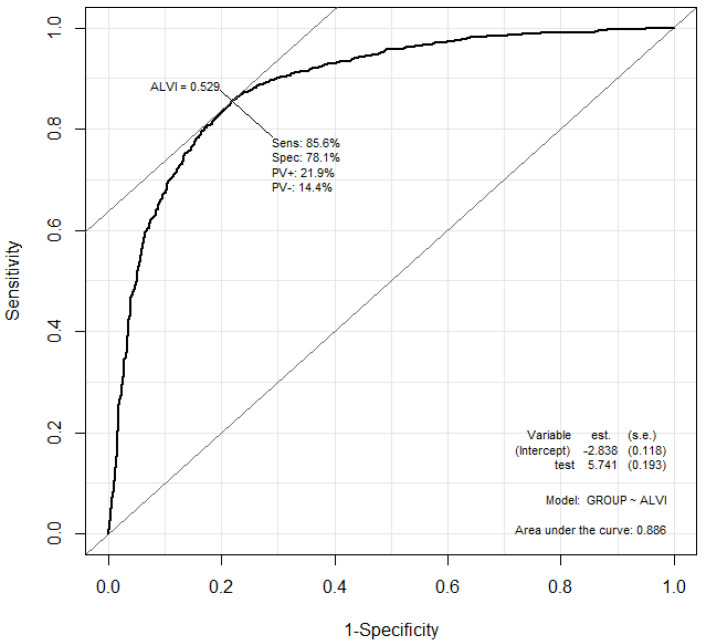
Receiver operating characteristic curve when arousal level voice index identifies utterance data of low arousal level and high arousal level in the Interactive Emotional Dyadic Motion Capture database. The horizontal and vertical axes represent 1-specificity (false positive rate) and sensitivity (positive rate), respectively. (Note. ALVI: arousal level voice index).

**Figure 3 sensors-20-05041-f003:**
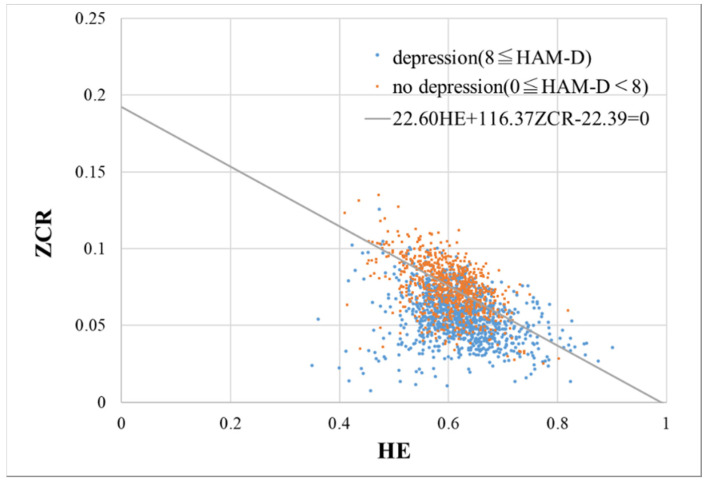
A scatter diagram of the Hurst exponent and zero crossing-rate calculated from each utterance (n = 1780) of depressed patients. The data of the no depression group and the depression group are shown in orange and blue, respectively. (Note. HAM-D: Hamilton Rating Scale for Depression; HE: Hurst exponent; ZCR: zero crossing rate).

**Figure 4 sensors-20-05041-f004:**
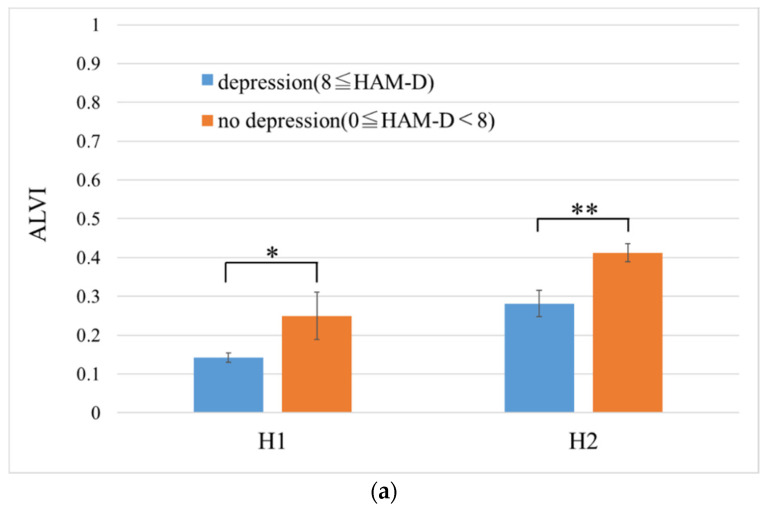
(**a**) The mean arousal level voice index for depression and no depression groups per hospital. Error bars represent standard error. (**b**) The mean HAM-D score for depression and no depression groups per hospital. Error bars represent standard deviation. *** (*p* < 0.001), ** (*p* < 0.01), * (*p* < 0.1). (Note. HAM-D: Hamilton Rating Scale for Depression; H1: Ginza Taimei Clinic; H2: National Defense Medical College; ALVI: arousal level voice index).

**Figure 5 sensors-20-05041-f005:**
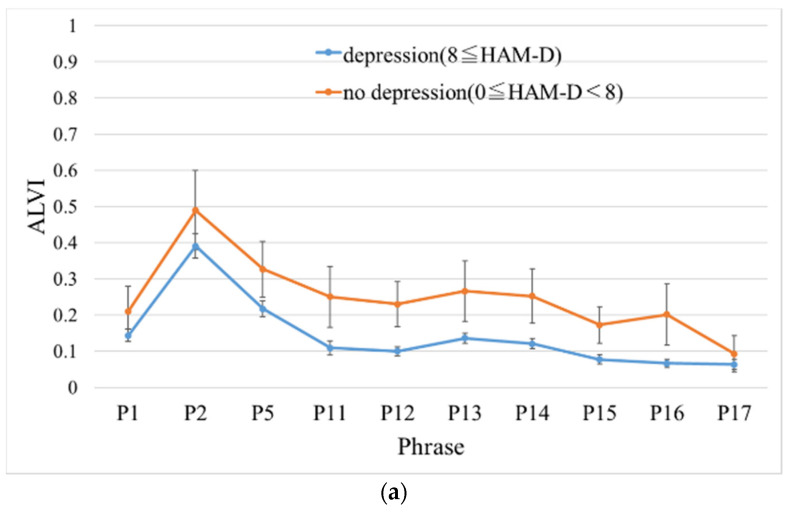
The mean of arousal level voice index of the no depression and depression groups for each phrase. (**a**) represents Ginza Taimei Clinic (H1) and (**b**) represents National Defense Medical College (H2). Error bars represent standard error. (Note. ALVI: arousal level voice index; HAM-D: Hamilton Rating Scale for Depression).

**Table 1 sensors-20-05041-t001:** Ten phrases used for recording.

Phrase	Phrase in Japanese	Purpose (Meaning)
P1	I-ro-ha-ni-ho-he-to	Non-emotional (no meaning, similar to “a-b-c”)
P2	Honjitsu ha seiten nari	Non-emotional (It is fine today)
P5	Mukashi aru tokoro ni	Non-emotional (Once upon a time, there lived)
P11	Garapagosu shotou	Check pronunciation (Galápagos Islands)
P12	Tsukarete guttari shiteimasu.	Emotional (I am tired/dead tired)
P13	Totemo genki desu	Emotional (I am very cheerful)
P14	Kinou ha yoku nemuremashita	Emotional (I was able to sleep well yesterday)
P15	Shokuyoku ga arimasu	Emotional (I have an appetite)
P16	Okorippoi desu	Emotional (I am irritable)
P17	Kokoroga odayaka desu	Emotional (My heart is calm)

**Table 2 sensors-20-05041-t002:** Participants’ information.

Hospital	Sex	Number of Subjects	Mean Age ± SD
H1	Female	55	31.6 ± 8.6
	Male	33	32.5 ± 6.5
	Total	88	32.0 ± 7.9
H2	Female	44	62.0 ± 13.1
	Male	46	48.8 ± 13.5
	Total	90	55.2 ± 14.8

Note: H1: Ginza Taimei Clinic; H2: National Defense Medical College; SD: Standard deviation.

**Table 3 sensors-20-05041-t003:** Mean scores on the Hamilton Rating Scale for Depression.

Hospital	Group	Number of Subjects	Mean HAM-D Score ± SD
H1	No depression (HAM-D < 8)	10	4.8 ± 1.3
	Depression (HAM-D ≧ 8)	78	24.4 ± 8.5
	Total	88	22.2 ± 10.1
H2	No depression (HAM-D < 8)	65	2.2 ± 2.2
	Depression (HAM-D ≧ 8)	25	15.3 ± 7.2
	Total	90	5.8 ± 7.2

Note. HAM-D: Hamilton Rating Scale for Depression; SD: Standard deviation.

**Table 4 sensors-20-05041-t004:** A summary of classification performance between the no depression and depression groups through arousal level voice index.

Phrase	*p*-Value ^a^		AUC	
	H1	H2	H1	H2
P1	0.33	0.027 *	0.60	0.65
P2	0.30	0.0092 **	0.60	0.68
P5	0.20	**0.00060** *******	0.63	**0.74**
P11	0.096 *	0.29	0.66	0.57
P12	**0.040** *****	0.016 *	**0.70**	0.67
P13	0.17	0.0047 **	0.63	0.69
P14	0.096*	0.0062 **	0.66	0.69
P15	0.099*	0.040 *	0.66	0.64
P16	0.19	0.022 *	0.63	0.68
P17	0.28	0.028 *	0.61	0.65

Note. H1: Ginza Taimei Clinic; H2: National Defense Medical College; AUC: Area under the curve. The minimum *p*-value and maximum AUC for each hospital are shown in bold type. *** (*p* < 0.001), ** (*p* < 0.01), * (*p* < 0.1). ^a^ By Wilcoxon rank sum test.

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
