# Peer review of "Evaluation of the Severity of Major Depression Using a Voice Index for Emotional Arousal"

_sensors, 2020, doi:10.3390/s20185041_

Round 1

Reviewer 1 Report

This is an excellent contribution to the literature in terms of the sound methods, analyses and sample. However, the introduction and discussion must be much improved on clarity and breadth (in terms of connecting analysis to depression literature). Some specific recommendations are below:

Line 48 "and self-administered questionnaires have reporting bias issues"- need a citation for this. -In which way- under reporting for men? Over-reporting for women with previous depression? Address duration issue? Perhaps speech is better marker as self-reports typically address the past 2 weeks. Speech can be more sensitive to changes over time.

Line 58- describe the Cannizzaro study results more.

Line 60- describe the Hashim results better

The paragraph between lines 57-72. Needs better organization. If all in the same paragraph, describe the results, speech features and analytic approach in order for each.

Arousal paragraph- describe the actual directional relationships between depression and markers of arousal. Are there different phases of arousal? First elevated, then dampened over time? What about first onset vs chronic depression? Duration? Why do authors even have a hypothesis based on arousal. Needs to be supported linearly and clearly. There is good evidence for this, it just must be summarized in the introduction.

Line 87 Please describe ALVI briefly here as it is mentioned in the abstract

144 describe qualitatively what is HE

Acronyms would be easier if they were labels H1 (hospital 1) and H2 instead of GTC and NDMC.

did you have any hypotheses about emotion vs neutral phrases (despite no found interactions?) why or why not?

The differences between hospitals is interesting (both with HAMD scores and voice analytics). Is this fully explained by demographic variables (age? poverty) - mention in discussion.

It might be helpful to overlay depressive symptoms by hospital onto the voice analytics by hospital to show that there are significant differences across the measures and not just in voice (more clearly).

The hospital differences highlight limitations for future use as universal screening. Please address what you can in analyses and discuss generalization issues in the discussion.

If the 2 groups were mild depression vs severe depression (not depression vs no depression), please explain them differently in the results/figures, or first state that they all have a history of depression in the introduction clearly as well. What are your speculations about how this may impact the data compared to depression vs controls.

Please add paragraph comparing and contrasting arousal findings to literature on depression and compare to other physiological markers.

Line 283. Revise for clarity/organization of this paragraph

Reviewer 2 Report

The manuscript presents a solid investigation of major depressive disorder using two vocal accoustic features as indices of arousal. The manuscript is logically sound and compelling. However, I have a few concerns:

  1. The authors imply the BDI instrument is not a self-report instrument (pg 1), this is not accurate. 
  2. The authors suggest that the present analyses don't require a dedicated device, but go on to use a dedicated microphone. This is misleading. The authors can suggest that future work examine whether the current investigation generalizes to a mobile setting or IOT device, but this conclusion in the introduction section is not warranted. 
  3. Paired raters were used to code arousal, nevertheless interrater reliability is not reported. This should be calculated and reported (suggested analysis format is intraclass correlations). 
  4. The authors state that they present the analysis scripts and data, but this is not linked within in the manuscript or elsewhere. Both the derived data and analysis scripts should be published as is stated. 

Reviewer 3 Report

Main message of the article

This article presents a novel method based on speech analysis to detect depression in adults.

General Judgment Comments

Overall, the article is well written and clear. The methods seem scientifically sound. All the methods are presented in detail, and with adequate in-text citations to previous works, but some more details would help increase the reproducibility of the work.

Suggestion: minor revision 

The article is overall of sufficient quality. I recommend the editor to accept the manuscript given that some revisions are made.

Major Issues

  • On line 42, it is reported that economic loss is the reason why you (and others in different fields) are working on non-invasive and objective systems to identify depression. But economic loss is not the only reason. Preventing self-harm, as well as harm towards others may be other reasons. I recommend the authors to make the sentence less exclusive, by either adding something on the line of “Amongst other reasons” or by adding more examples.
  • On line 58. Cannizzaro’s study is reported. To better understand the importance and similarities (and differences) of the present work, I suggest the authors to add an indication of the features that were used in Cannizzaro’s work after “voice acoustics” (Line 59). The same applies to Line 68.
  • On line 107 it is reported that 17 phrases have been collected, but only 10 have been analyzed. However, one may wonder whether there is an effect of the order of reading? Were participants instructed to read the sentences in the same order? If this is the case, the authors should address this in the limitations of the study. If not, would it be possible to test whether the order of pronunciation had any effect? I believe this is a crucial criticality of the employed method.
  • Line 194. As noted in previous works on the analysis of speeches and other human vocalizations, there are some critical steps in studies that analyze voice. One is the preprocessing used before the analysis of the signals (e.g. Gabrieliet al. 2019. Are cry studies replicable? An analysis of participants, procedures, and methods adopted and reported in studies of infant cries. Acoustics). For example, it is not clear from your work whether a filter has been used to remove noise components from audio recordings prior to the estimation of the HE and ZCR, also given that on line 222 is reported that “Likewise, the sound field environment may also vary”. Can the authors please clarify on this point?
  • Equation 1. Continuing on the previous point, it is not clear to me how were HE and ZCR extracted. Has the analysis been conducted on the raw signals or was an audio feature extraction stage employed, given that in the introduction studies with features extractions are cited? For example, what is x in Equation 1? Is it the amplitude of the signal? Please clarify what is the independent variable of both the equations.
  • Line 194. Here it is reported that R has been used for the analysis. Given that there are multiple versions of R available, and that some of the packages / functions are OS-dependent, I would suggest the authors to report the version of R-core and the version of the packages used, as well as the Operating System on which the analysis has been performed, as noted in Gabrieli, G., Scapin, G., Bornstein, M. H., & Esposito, G. (2019, December). Are cry studies replicable? An analysis of participants, procedures, and methods adopted and reported in studies of infant cries. In Acoustics (Vol. 1, No. 4, pp. 866-883). Multidisciplinary Digital Publishing Institute.

Minor Issues

  • Line 84. If i got this correctly, your proposed algorithm detects depression using a voice index based on the estimation of the arousal. If this is the case, I would rephrase this as ”In this paper, we examined the possibility of detecting depression using a voice index based on the estimation of speech’s  arousal level.” for clarity.
  • Line 288. “However, it may be advantageous because the ALVI is unlikely to be overfitted. After all, it consists of only two indices, HE and ZCR.” I am quite confident one can achieve overfitting even with as little as two variables.  
  • Table 4. Given the huge number of tests, one would assume that a correction to the p-value has been performed. Is this the case? If not, I would suggest to use the term “uncorrected p-value” for better clarity.
  • Line 50, it may be worth also reporting studies that investigate speeches and biomarkers of an individual B to detect depression in an individual A, which is a special case of non-invasive and objective technique to detect depression. See for example Gabrieli, G., Bornstein, M. H., Manian, N., & Esposito, G. (2020). Assessing Mothers’ Postpartum Depression From Their Infants’ Cry Vocalizations. Behavioral Sciences, 10(2), 55 or Grace, S.L.; Evindar, A.; Stewart, D. The effect of postpartum depression on child cognitive development and behavior: A review and critical analysis of the literature.Arch. Women’s Ment. Health 2003.

Final comments

The manuscript is clear, interesting, and well written, however, some minor edits may help improve the quality and reproducibility of the work.
